# The Analysis of Mechanical Structure of a Robotic Leg in Running for Impact Mitigation

**Jungsoo Cho [1],[†]** and **Kyoungchul Kong [2],***

[1]  Department of Mechanical Engineering, Sogang University, Seoul 04107, Korea; juchowiwang13@hanmail.net
[2]  Department of Mechanical Engineering, Korea Advanced Institute of Science and Technology, Deajeon 34141, Korea
*  Correspondence: kckong@kaist.ac.kr
†  Current address: 35, Baekbum-ro, Mapo-gu, Seoul 04107, Korea.

**Abstract:** Legged robots suffer from the impact due to the consistent collisions with the ground. At the moment of collision, the sudden impact force not only causes the legs to lose contact off the ground, but can also reduce controllability and durability. This phenomenon becomes worse for the robots in running. In order to mitigate such an impact effectively, this study focuses on the mechanical structure of the legs, unlike the previous studies, which focused on the component level. The mechanical structures include actuator configuration, segment ratio, total length, and flexion direction. Contact inertia (CI), closely related to the impact, is derived and utilized to analyze the mechanical structure in terms of impact mitigation. A series of impact experiments with a fabricated leg verify that the mechanical structure affects mitigating the impact.

**Keywords:** impact; impulse; contact inertia; mechanical structure; running; legged robot

## 1. Introduction

Legged robots with high versatility have been intensively studied and developed to replace humans in hazardous or tedious and repetitive tasks by many research teams. Boston Dynamics developed "SpotMini", which can navigate cramped areas like construction sites autonomously, climbing steps and avoiding obstacles [1]. MIT recently presented the legged robots by scale. "MIT Cheetah III" shows a broad range of leg movement and more focus on versatility compared to its previous version [2], and "MIT Mini Cheetah", a miniature version of "MIT Cheetah III", is capable of acrobatic motions like back-flip in addition to all capabilities of its predecessor [3]. UCLA devised a different usage of legs by taking advantage of legs and feet as arms and hands [4]. Many other research teams successfully built their own legged robot platforms with high versatility [5–8]. Meanwhile, legged robots have been branched out for high-speed locomotion. "WildCat", regarded as the fastest quadrupedal robot in history, can run as fast as 8.4 m/s [9]. "MIT Cheetah II" showed a remarkable performance of jumping over several obstacles in the middle of running [10]. "Raptor", a biped robot, showed the highest speed locomotion, as fast as 12.7 m/s, though tethered on a treadmill [11,12].

Most legged robots mentioned in the previous paragraph have their biological counterparts such as dogs, cats, or cheetahs. As their speed increases and they begin to run, their dynamics is completely different from when they walk. The locomotion of legged animals is typically divided into two phases: stance and swing. When they run, another version of the swing phase, i.e., no feet in contact with the ground, rises contrary to the original version of the swing phase, i.e., at least one foot on the ground, which is observed during walking [13–15]. At the end of the swing phase, a foot or feet touch(es) the ground. Due to the collision with the ground, an impact force is suddenly transferred

to the foot, to the leg, and then to the body, in that order. Fortunately, periarticular soft tissues and sponge-like cancellous (or trabecular) bones, located on both ends of the bone, attenuate the peak forces [16,17]. Since legged robots structured mostly with rigid bodies are vulnerable to the peak forces in collision, however, the foot momentarily loses contact [18,19], which would severely deteriorate the controllability of stance, as in Figure 1, showing our previous hopping experiment. In the hopping experiment, virtual compliance control was applied to the leg, but as soon as the tip of the leg touched the ground, a major collision occurred at the beginning, followed by a few minor collisions, as shown in Figure 1a. This phenomenon prevented the leg from being compressed enough to bounce back to the air. The virtual compliance was adjusted to mitigate the impact, but a similar phenomenon was repeated. The leg was even fractured after a few sets of experiments, as shown in Figure 1b.

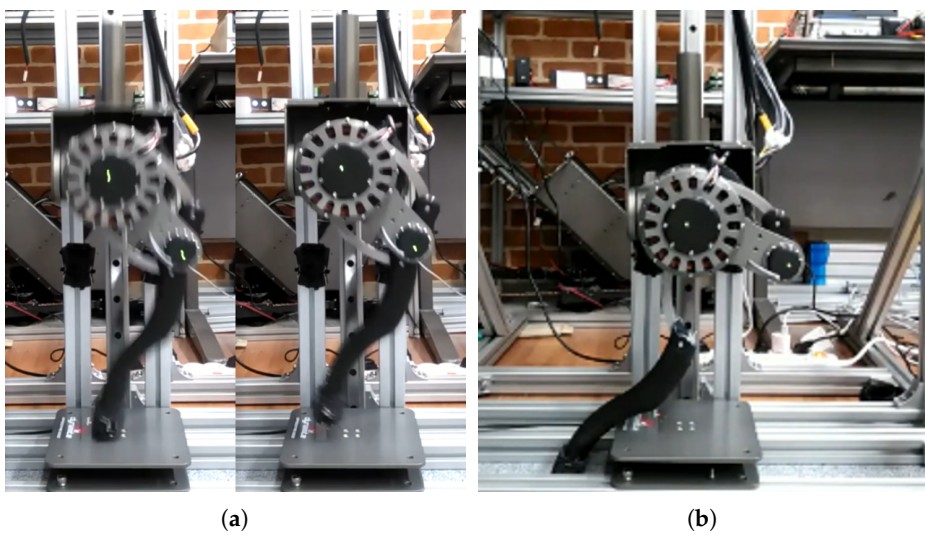

(**a**)             (**b**)

**Figure 1.** Effect of sudden impact: (**a**) contact loss; (**b**) fracture.

There have been dedicated efforts to attenuate such ground impact. Serial-elastic actuators (SEAs) were adopted in "ANYmal", developed by ETH [5], which gave mechanical compliance between the gearbox output and a joint [20–22], or springs were applied to the foot such that the robot could be adapted to dynamic interactions directly with the ground [23]. For "MIT Cheetah II", MIT pursued the "massless" concept ideal for impact mitigation with light legs and a low gear ratio. The legs featured the maximized strength-to-weight ratio through a tendon-bone mechanism inspired by biology [24]. Adding on to those light legs, the low gear ratio enabled high-bandwidth force control through direct contact without any medium such as mechanical compliance [25]. As previously mentioned, the current research on impact mitigation is limited to component-level approaches covering the mechanical components that make up the leg such as actuators, including the gearbox, linkages, and compliant mechanisms like springs and dampers. Therefore, the approach needs to be leveled up to cover the structural part of the leg.

In order to mitigate the impact at landing of legged robots in running, this paper deals with a variety of mechanical structures such as actuator configuration, segment ratio, and overall length and flexion direction. First of all, the impulse at the point of contact is defined to quantify the impact. The impulse is expressed as the product of an impact velocity and contact inertia (CI), where CI is closely related to the mechanical structure, as well as the mechanical components. CI, which can be derived from dynamics, starts with single rigid body dynamics and then extends to the dynamics of articulating two rigid bodies that can represent a leg. CIs are compared by the mechanical structure listed above through simulations. In order to verify that the mechanical structure affects mitigating the impact, a leg is fabricated and a few sets of experiments are implemented for different mechanical structures, as well as for different attack angles and impact directions. The measured impact forces in

the experiments are transformed to CIs, and finally, CIs are evaluated by comparing to CIs simulated from the dynamics.

## 2. Methods

### 2.1. Impulse

In this section, the impact led by the tip of a leg that contacts the ground, falling from the air, is quantified by a metric called "impulse". The object of the impulse first starts with a point-mass and expands to a single rigid body.

#### 2.1.1. Point-Mass

Imagine a point-mass colliding vertically with the ground as shown in Figure 2a. The point-mass, starting to fall from at some point in the air, is accelerated to the ground, and the velocity gradually increases. At the very moment when it reaches the ground and the velocity suddenly turns to zero, the impulse of the point-mass can be defined as:

$$I = -m\dot{x}, \tag{1}$$

where $m$ and $\dot{x}$ denote the mass and velocity right before the collision, respectively. The impulse equals the change of momentum as in (1), which implies that the impulse increases as the mass and colliding velocity increase. More generally, the term of $m$ in (1) represents an inertia at the contact against the impact force, i.e., $f$. Subsequently, the impulse will be dealt with in more detail for a single rigid body.

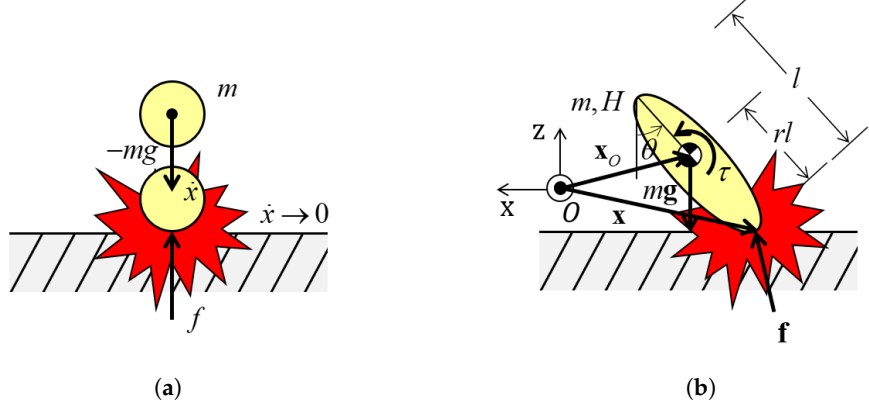

**Figure 2.** Impulse: (**a**) a point-mass; (**b**) a single rigid body.

#### 2.1.2. Single Rigid Body

Let the colliding object expand from a point-mass to a single rigid body more realistic in our physical world. Likewise, a rigid body falls from the air. The difference with a point-mass is that a rigid body has a volume such that it can rotate by an angle of $\theta$, as shown in Figure 2b. To define the impulse of a rigid body, it should be started from the impact dynamics of a single rigid body. Letting $\mathbf{q} = \begin{bmatrix} \mathbf{x}_O & \theta \end{bmatrix}^\top \in \mathbb{R}^{3\times1}$, the impact dynamics is described as:

$$\mathbf{H}\ddot{\mathbf{q}} + \mathbf{G} = \mathbf{T}, \tag{2}$$

where $\mathbf{H} \in \mathbb{R}^{3 \times 3}$, $\mathbf{G} \in \mathbb{R}^{3 \times 1}$, and $\mathbf{T} \in \mathbb{R}^{3 \times 1}$ denote the inertia, gravity, and external force matrices, respectively:

$$\mathbf{H} = \begin{bmatrix} m\mathbf{I} & \mathbf{0} \\ \mathbf{0}^{\top} & H \end{bmatrix}, \quad \mathbf{G} = \begin{bmatrix} m\mathbf{g} \\ 0 \end{bmatrix}, \quad \text{and} \quad \mathbf{T} = \begin{bmatrix} \mathbf{f} \\ \tau \end{bmatrix} ; \tag{3}$$

where $m$, $H$, and $\tau$ denote the mass and inertia of a rigid body and external torque, respectively; where $\mathbf{I} \in \mathbb{R}^{2 \times 2}$, $\mathbf{0} \in \mathbb{R}^{2 \times 1}$, $\mathbf{g} \in \mathbb{R}^{2 \times 1}$, and $\mathbf{f} \in \mathbb{R}^{2 \times 1}$ denote an identity matrix and null, gravity acceleration, and impact force vectors, respectively.

The impact dynamics of (2) in terms of the general coordinates, i.e., $\mathbf{q}$, should be transferred in terms of the contact coordinates, i.e., $\mathbf{x} \in \mathbb{R}^{2 \times 1}$, against the impact force, i.e., $\mathbf{f}$. Let:

$$\mathbf{J} = \begin{bmatrix} \mathbf{I} & \frac{\partial \mathbf{R}(\theta)}{\partial \theta} r\hat{l} \end{bmatrix} \in \mathbb{R}^{2 \times 3}, \tag{4}$$

where $\mathbf{R} \in \mathbb{R}^{2 \times 2}$ is a rotation matrix with an angle of $\theta$; $r \in (0, 1)$ is the center of mass (COM); $\hat{l} = \begin{bmatrix} 0 & -l \end{bmatrix}^{\top}$, where $l$ is the length of rigid body. Note that if COM moves to the contact point, $r$ goes to 0. Then, $\mathbf{q}$ and $\mathbf{T}$ can be converted by (4) in terms of $\mathbf{x}$ and $\mathbf{f}$:

$$\dot{\mathbf{q}} = \mathbf{J}^{-1}\dot{\mathbf{x}} \quad \text{and} \quad \mathbf{T} = \mathbf{J}^{\top}\mathbf{f} . \tag{5}$$

Applying (5) to (2), the impact dynamics in terms of $\mathbf{x}$ is finally derived as:

$$\mathbf{\Lambda}\ddot{\mathbf{x}} + \mathbf{h} = \mathbf{f}, \tag{6}$$

where:

$$\mathbf{\Lambda} = \left( \mathbf{J}\mathbf{H}^{-1}\mathbf{J}^{\top} \right)^{-1} \quad \text{and} \quad \mathbf{h} = \mathbf{J}^{-\top}\mathbf{G} - \mathbf{\Lambda}\dot{\mathbf{J}}\dot{\mathbf{q}}. \tag{7}$$

In the same manner as (1), the impulse of single rigid body can be expressed as:

$$I = \|\mathbf{\Lambda}\dot{\mathbf{x}}\| , \tag{8}$$

where $\mathbf{\Lambda}$ is called a contact inertia matrix (CIM). The CIM of a single rigid body is as follows:

$$\mathbf{\Lambda} = \frac{m}{1 + \frac{mr^2 l^2}{H}} \begin{bmatrix} 1 + \frac{mr^2 l^2}{H} sin^2\theta & \frac{mr^2 l^2}{H} cos\theta sin\theta \\ \frac{mr^2 l^2}{H} cos\theta sin\theta & 1 + \frac{mr^2 l^2}{H} cos^2\theta \end{bmatrix} . \tag{9}$$

For an impact direction, i.e., $\mathbf{v} = \dot{\mathbf{x}}/\|\dot{\mathbf{x}}\|$, contact inertia (CI) can be defined as:

$$\lambda = \|\mathbf{\Lambda}\mathbf{v}\| . \tag{10}$$

Therefore, the impulse is determined by CI in (10) for a given impact speed of $\|\dot{\mathbf{x}}\|$.

### 2.2. Contact Inertia: Single Rigid Body

In this section, the effect of parameters on CI, i.e., $\theta$ and $r$ in (9), is investigated through a case study and verified through a simple experiment.

### 2.2.1. Case Study

Suppose two extreme cases: a single rigid body drops straight down to the ground (i) with $\theta = 0$ and (ii) with $\theta = -\pi/2$, as shown in Figure 3. Note that the contact points of both cases are assumed

to be equal. Since only gravity works in the both cases, the impact direction is the negative $z$-direction, i.e., $\mathbf{v} = \begin{bmatrix} 0 & -1 \end{bmatrix}^{\top}$. Then, CIs denoted by $\lambda_1$ and $\lambda_2$ for each case are calculated as follows:

$$\lambda_1 = m \quad \text{and} \quad \lambda_2 = \frac{m}{1 + \frac{mr^2 l^2}{H}}. \tag{11}$$

As can be seen from (11), CI varies on body angle, i.e., $\theta$. Unlike the first case of $\theta = 0$, where CI is exactly the same as in the case of a point-mass, CI in the second case of $\theta = -\pi/2$ must be equal to or smaller than the first case, varying on $r$ and $H$, which are dependent on each other, provided $m$ and $l$ are given.

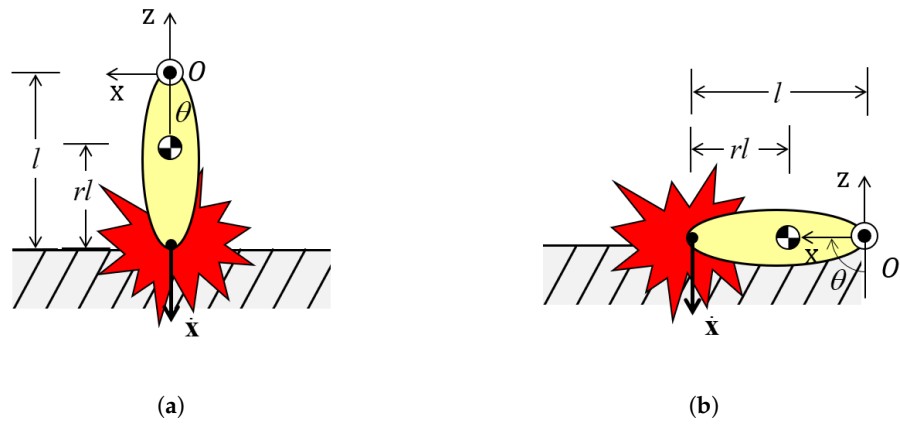

(a)　　　　　　　　　　　　　　　　　　　　　　(b)

**Figure 3.** Two extreme cases: (**a**) $\theta = 0$; (**b**) $\theta = -\pi/2$.

### 2.2.2. Center of Mass

More attention is required in the second case. Since COM, i.e., $r$, is a function of mass distribution, a density function of $t \in [0, l]$ is defined for thorough investigation of $\lambda_2$ in (11):

$$\rho(t) = \frac{m}{l}(n+1)t^n, \tag{12}$$

where $n \in (-1, \infty)$. It can be known from (12) that the integral of $\rho(t)$ over $[0, l]$ becomes $m$. If $n$ goes to $-1$, the mass is dense at the tip of contact. If $n$ goes to infinity, on the contrary, the mass is dense at the proximal part. Given (12), COM, i.e., $r$, and the inertia of a rigid body about $O$, i.e., $H_O$, can be calculated as follows:

$$r = \frac{\int_0^l t \, dm}{m} = \frac{n+1}{n+2}l \tag{13}$$

and:

$$H_O = \int_0^l t^2 dm = \frac{n+1}{n+3}ml^2. \tag{14}$$

According to the parallel axis theorem, the inertia about $r$, i.e., $H$, can be calculated using (13) and (14).

$$H = H_O - mr^2. \tag{15}$$

Applying (13) and (15), $\lambda_2$ can be calculated depending on $r$ as shown in Figure 4. As COM drifts away from the point of contact, CI decreases.

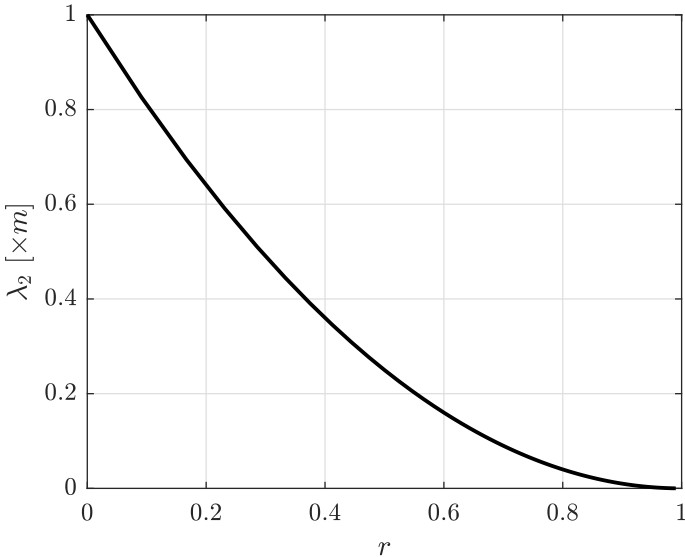

**Figure 4.** CI of a rigid body varying COM.

### 2.2.3. Linkage Drop Test

According to the previous case study, a rigid body dropping with an angle is expected to lessen the impulse compared to that without an angle. To demonstrate this, a simple linkage drop test was prepared. As the impulse is an integral of the impact force over the impact time, the impulse was difficult to measure directly in the experiment. However, the impact time duration was determined by the contact stiffness between the two objects colliding with each other, and the contact stiffness between the tip of the linkage and the ground was expected to be constant in the experiment. Therefore, the impulse could be considered the impact force when the linkage hit the ground.

The experiment proceeded in two directions with the linkage a part of the leg: one dropping the linkage without an angle, i.e., perpendicular to the ground, as in Figure 5a, and the other dropping it with an angle of about 25°, as in Figure 5b. To measure the impact force, a force plate (Type 9260AA, *Kistler Group*) was used. On the force plate, the linkage weighing 440 g dropped 20 times repeatedly from 0.33 m high to maintain the impact velocity, i.e., $\dot{x}$, as shown in Figure 5. Note that the tip of the linkage was covered with rubber to dampen the impact such that the impact minimized the damage to the sensor as much as possible. While repeatedly dropping the linkage, the first peak forces were logged. Then, they were averaged and the standard deviation taken, as shown in Figure 6. On average, the peak forces caused by the linkage dropping without an angle were higher than those with an angle. Thus, in line with (11), the experimental result confirmed that the impulse varied on CI, which varied on the angle of an object, i.e., $\theta$. Subsequently, the single rigid body expanded into an articulated rigid body, similar to the structure of a leg, for further investigation of the impact.

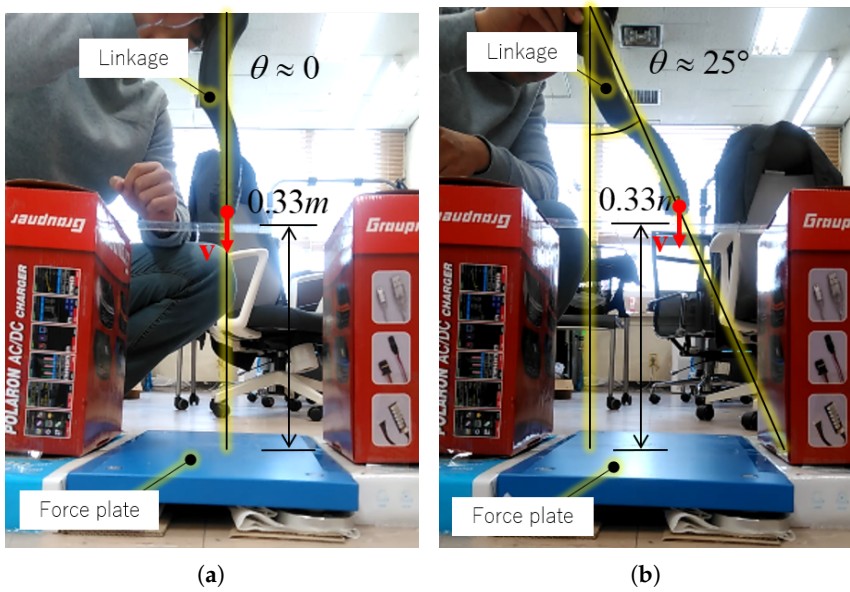

|  |  |
|:---:|:---:|
| (**a**) | (**b**) |

**Figure 5.** Linkage drop test: (**a**) with no angle; (**b**) with an angle.

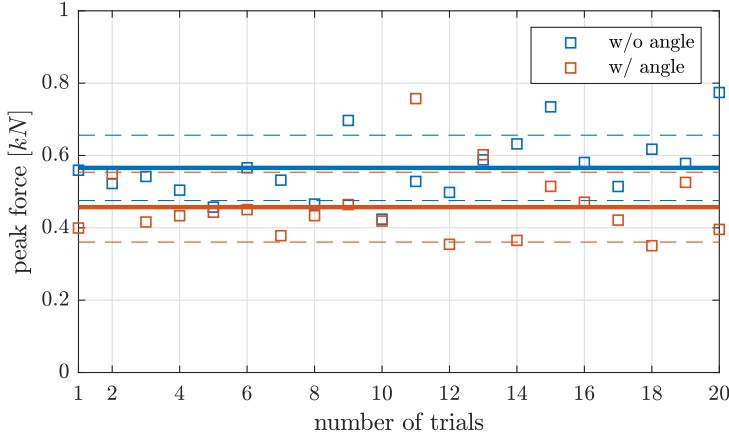

**Figure 6.** Linkage drop test result.

### 2.3. Contact Inertia: Articulated Rigid Body

#### 2.3.1. Impact Dynamics

In order to investigate the impact characteristics of the articulated rigid body, CIM, i.e., $\Lambda$ in (9), should be derived first. To this end, the dynamics of an articulated rigid body comprised of a body and upper and lower limbs can be expressed as follows:

$$\mathbf{H}(\mathbf{q})\ddot{\mathbf{q}} + \mathbf{B}(\mathbf{q}, \dot{\mathbf{q}}) + \mathbf{G}(\mathbf{q}) = \mathbf{T}, \tag{16}$$

where $\mathbf{H} \in \mathbb{R}^{4 \times 4}$, $\mathbf{B} \in \mathbb{R}^{4 \times 1}$, $\mathbf{G} \in \mathbb{R}^{4 \times 1}$, and $\mathbf{T} \in \mathbb{R}^{4 \times 1}$ denote the inertia, internal force, gravity, and external force matrices, respectively. It can be transformed from (16) to the impact dynamics in a similar way to Section 2.1.2:

$$\tilde{\mathbf{\Lambda}}(\mathbf{q})\ddot{\mathbf{x}} + \tilde{\mathbf{B}}(\mathbf{q}, \dot{\mathbf{q}}) + \tilde{\mathbf{G}}(\mathbf{q}) = \mathbf{f}. \tag{17}$$

The detailed derivation of the impact dynamics and detailed description of each term are elaborated in Appendix A.1. As can be seen in (17), the impact force of $\mathbf{f}$ is affected by the three terms. Since at the impact, the tip velocity of $\dot{\mathbf{x}}$ goes to zero instantaneously and the joint velocity

of $\dot{\mathbf{q}}$ also goes to zero, the effect of the second term related to $\dot{\mathbf{q}}$ can be ignored. The third term is so related to the gravity that it becomes dominant in steady states. On the contrary, the first term becomes dominant in transient states such as impacts.

### 2.3.2. Contact Inertia Ellipse

In (17), CIM of the articulated rigid body is defined as:

$$\tilde{\mathbf{\Lambda}} = \left[ \tilde{\mathbf{J}}(\mathbf{q}) \mathbf{H}^{-1} \tilde{\mathbf{J}}(\mathbf{q})^\top \right]^{-1} \in \mathbb{R}^{4\times4}, \tag{18}$$

where $\mathbf{H}$ consists of four sub-matrices as described in (A1): $\mathbf{H}_{11}$ $\mathbf{H}_{12}$ $\mathbf{H}_{21}$, and $\mathbf{H}_{22}$. According to Appendix A.2, the $\mathbf{H}_{11}$ of them is much larger than any other sub-matrix such as $\mathbf{H}_{12}$, $\mathbf{H}_{21}$, and $\mathbf{H}_{22}$ due to including the inertia to the body. Therefore, (18) can be reorganized as:

$$\mathbf{\Lambda} = \left[ \mathbf{J}(\mathbf{q}) \mathbf{H}_{22}^{-1} \mathbf{J}(\mathbf{q})^\top \right]^{-1} \in \mathbb{R}^{2\times2}. \tag{19}$$

As can be seen from (19), the inertia for the upper and lower limbs, i.e., $\mathbf{H}_{22}$, remained alone.

Now, CI of articulated rigid body can be defined in the same manner as in (10). For all impact directions of $\mathbf{v}$, which can be expressed as a unit vector that rotates $360°$, the CI draws an ellipse around the point of contact of the articulated rigid body. The ellipse is called the contact inertia ellipse (CIE). Given the impact speed of $\|\dot{\mathbf{x}}\|$, CIE shows how large the impact would be for an impact direction. Since in running situations, the impact velocity is hard to predict or measure, CIE is more useful in analyzing the impact than CI, which corresponds to a particular impact direction.

As can be seen from (19), CI is closely related to the Jacobian, i.e., $\mathbf{J}$. Since $\mathbf{J}$ involves the kinematics related to the mechanical structures and configurations of a leg that can be represented by an articulated rigid body, CIE is expected to be affected by actuator configuration, segment ratio, overall length, and flexion direction, which indicate the mechanical structures, and also affected by extension length and attack angle, as shown in Section 2.2.3, which indicate the configurations. In the Simulations section, therefore, CIEs are compared on the extension length and attack angle by the mechanical structure.

## 3. Simulations

Before the CIE simulation on mechanical structures, two fundamental assumptions were made: (i) all actuators were concentrated on the body, and (ii) the mass density of limbs was constant over the length. The first assumption is generally accepted. Since actuators have a relatively large mass due to gearboxes and motors compared to other mechanisms, it is desirable that they are far from the point of impact, as clearly shown in Section 2.2.2. The other is to maintain generality such that the mass increases with the length of the limb.

### 3.1. Actuator Configuration

The actuator configuration can be divided into serial and parallel configurations, as shown in Figure 7. The serial configuration takes the structure of hip and knee actuators connected in series as in Figure 7a such that the torque generated by the knee actuator affects the torque generated by the hip actuator while the motion generated by the knee actuator is affected by the motion generated by the hip actuator. The parallel configuration, on the other hand, features a structure different from the serial configuration. Hip and knee actuators are mounted on the body to independently operate the upper and lower limbs as in Figure 7b. For this reason, the motions and torques generated by the hip and knee actuators are not affected by each other.

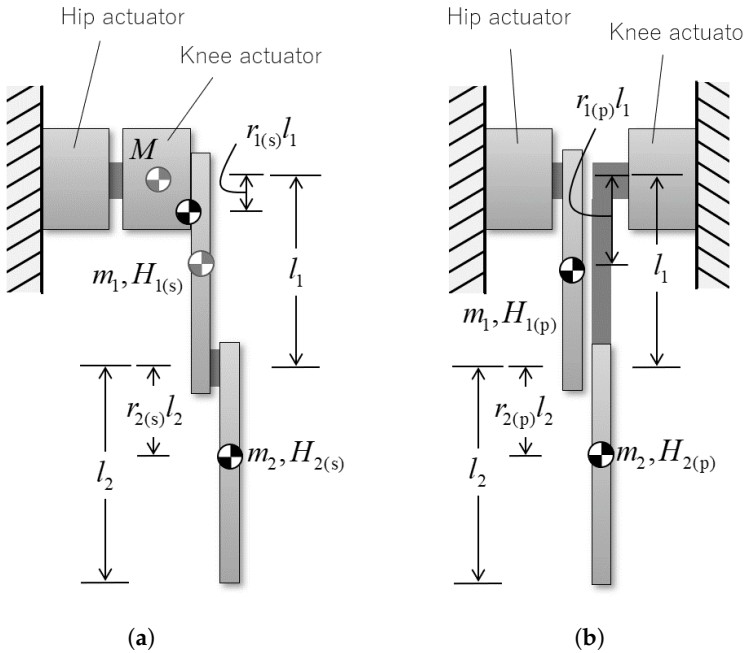

**Figure 7.** Actuator configuration: (**a**) serial; (**b**) parallel.

To simulate CIEs of each actuator configuration, the moments of inertia and COMs for each limb should be provided. According to the second assumption, the moments of inertia of upper and lower limb in parallel configuration are:

$$H_{1(p)} = \tfrac{1}{12} m_1 l_1^2, \quad H_{2(p)} = \tfrac{1}{12} m_2 l_2^2 \ , \tag{20}$$

and COM of upper and lower limb is $r_{1(p)} = r_{2(p)} = 0.5$. In serial configuration, the mass of the knee actuator, i.e., $M$, along with the mass of the link, i.e., $m_1$, is included in the mass of the upper limb. Therefore, the moment of inertia and COM for the upper limb are:

$$H_{1(s)} = \left[ \frac{1}{12} + \left( \frac{1}{2} - r_1 \right)^2 \right] m_1 l_1^2 + M r_1^2 l_1^2 \tag{21}$$

and:

$$r_{1(s)} = \frac{m_1}{2 \left( m_1 + M \right)}. \tag{22}$$

The moment of inertia and COM for the lower limb are equal to the parallel configuration, i.e., $H_{2(s)} = H_{2(p)} = H_2$ and $r_{2(s)} = r_{2(p)} = r_2$. Using the moments of inertia and COMs, CIEs of each actuator configuration could be simulated by (19). Note that the Jacobian matrix of the parallel configuration is different from (A6) of the serial configuration, i.e.,

$$\mathbf{J}(\mathbf{q}) = \begin{bmatrix} \frac{\partial \mathbf{R}(\theta_1)}{\partial \theta_1} \hat{l}_1 & \frac{\partial \mathbf{R}(\theta_{12})}{\partial \theta_{12}} \hat{l}_2 \end{bmatrix} \in \mathbb{R}^{2 \times 2}, \tag{23}$$

where $\theta_1$ and $\theta_2$ denote the relative joint angles of upper and lower limbs, respectively, as shown in Figure A1; $\theta_{12}$ is abbreviated as $\theta_1 + \theta_2$. $\hat{l}_1$ and $\hat{l}_2$ represent neutral length vectors of the upper and lower limbs in the same manner as in $\hat{l}$ of (4).

CIEs of both actuator configurations were exactly the same with regard to $l_n$ and $\theta$, as shown in Figure 8. $l_n$ and $\theta$ denote the extension ratio and attack angle, respectively. When the leg fully extended, for example, $l_n$ became one. The reason why both CIEs of serial and parallel configurations resulted in being the same is because both moments of inertia about the upper joint axis for the upper

limb were consequently the same. It can be interpreted that the mass of the knee actuator, i.e., $M$, made COM drift away from the point of contact, which eventually compensated the increased mass of the upper limb in the serial configuration. In this regard, a detailed mathematical proof is given in Appendix B.

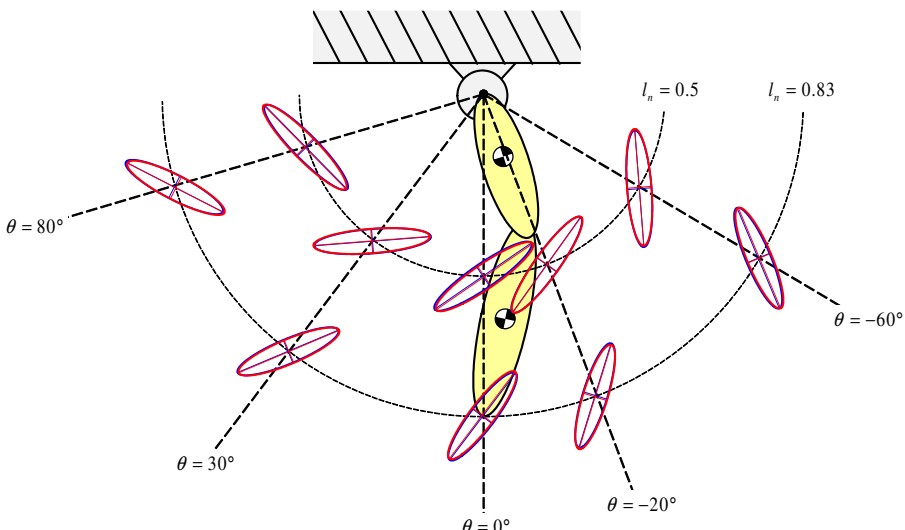

**Figure 8.** CIEs depending on the actuator configuration: serial (red) and parallel (blue).

### 3.2. Segment Ratio

To investigate the effect of the segment ratio of the upper and lower limb, an additional assumption should be taken besides those previously mentioned: (iii) a total length of the leg, i.e., $l_1 + l_2$, remained constant, which constrains the effect of the total length change. The segment ratio is defined as a length ratio of the upper limb to lower limb, i.e., $r = l_1/l_2$. CIEs depending on $r$ were simulated by (19), as shown in Figure 9. As $r$ increased, i.e., the length of upper limb was relatively longer than that of lower limb, the CIE spread thin, which implies that the impact varies drastically depending on the impact direction. On the other hand, as $r$ decreased, the CIE swelled, and thus so would be the impact.

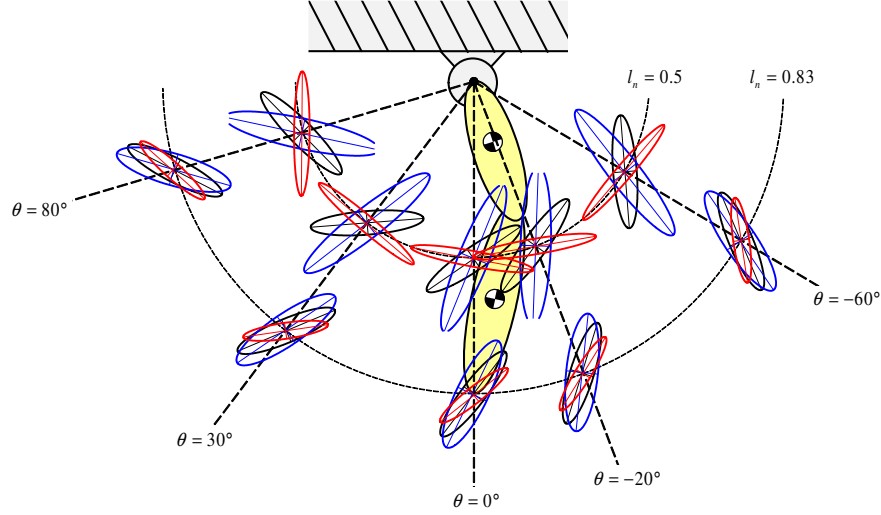

**Figure 9.** CIEs depending on a segment ratio: $r > 1$ (red), $r < 1$ (blue), and $r = 1$ (black).

### 3.3. Overall Length

While the segment ratio was constrained as $r = 1$, the case where the total leg length lengthened or shortened was taken into account. If the total length lengthens, it benefits an efficient tip velocity with respect to joint velocities, which in turn leads to a farther stride. However, as the mass increased with the length, and the moment of inertia increased with the mass, and CIE expanded, as shown in Figure 10.

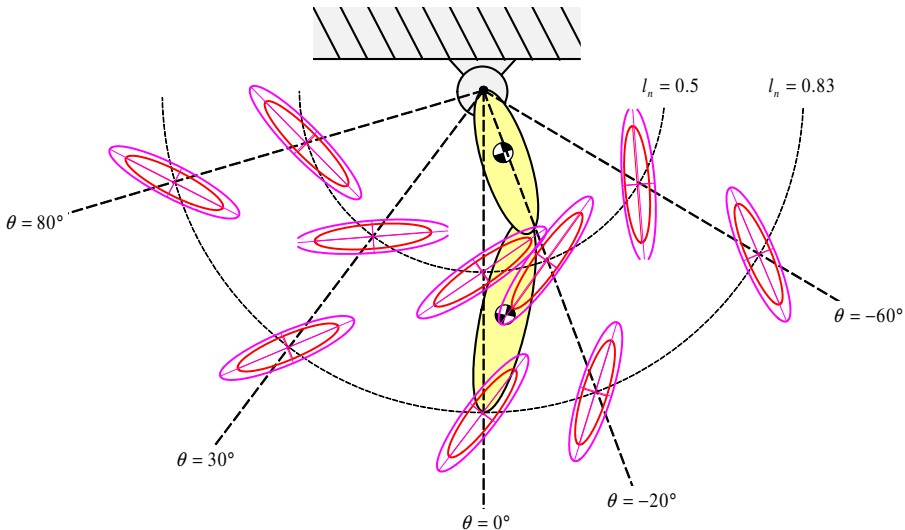

**Figure 10.** CIEs depending on overall length: $l_1 + l_2$ (pink) $> l_1 + l_2$ (red).

### 3.4. Flexion Direction

As so far shown, CIEs appeared as rotated at some angle, which means that CI varies on the impact direction, i.e., the impact on the leg varies on the impact direction. Although the impact direction is difficult to predict, the possible range of impact directions can be limited between $\mathbf{v}_H$ and $\mathbf{v}_D$ as shown in Figure 11, taking into account the running situation. $\mathbf{v}_H$ and $\mathbf{v}_D$ represent the heading direction vector induced by the body's inertia and the downward direction induced by gravity, respectively. $\mathbf{v}_D$ is invariant, but $\mathbf{v}_H$ depends on which direction the robot was running. In terms of mechanical structure, the robot's heading direction can be converted to the leg's flexion direction. For example, a situation where a robot is running backward with its legs flexed forward can be considered exactly the same as the situation where the robot is running forward with its legs flexed backward. Therefore, the impact could vary on the flexion direction of a leg as well as the actuator configuration, segment ratio, and overall length.

Let $\lambda_B$ be CI of a leg running backwardly flexed:

$$\lambda_B = \|\Lambda_B \mathbf{v}_1\|, \tag{24}$$

where $\Lambda_B$ represents a CIM determined by given mechanical structures including the backward flexion and $\mathbf{v}_1$ denotes an impact direction. Then, CI of the leg running forwardly flexed for the same impact direction, i.e., $\lambda_F$, can be defined as:

$$\lambda_F = \|\Lambda_F \mathbf{v}_1\|, \tag{25}$$

where $\Lambda_F = \mathbf{S}^\top \mathbf{R}^\top (-2\theta) \Lambda_B \mathbf{R}(-2\theta) \mathbf{S}$. $\mathbf{S}$ denotes a transformation matrix that reverses only the heading direction of $\mathbf{v}_1$, i.e.,

$$\mathbf{S} = \begin{bmatrix} -1 & 0 \\ 0 & 1 \end{bmatrix}. \tag{26}$$

Therefore, CIE depending on the flexion direction was simulated in a way of décalcomanie as shown on the right side of Figure 11, where CIE with a gray dashed line was folded in half about the z-axis of the leg into a gray solid line.

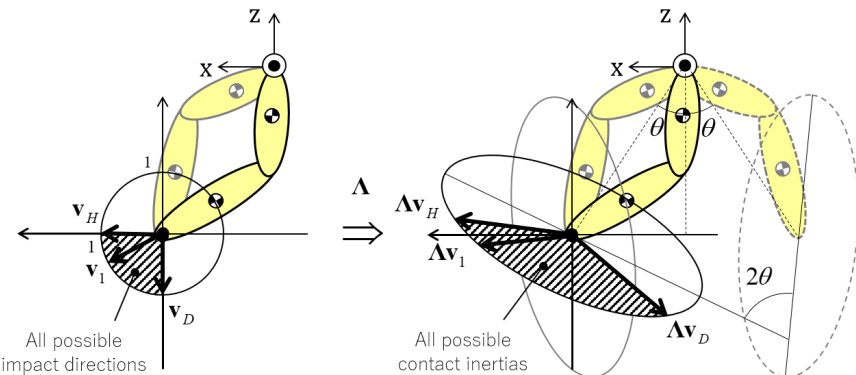

**Figure 11.** CI for all possible impact directions and CIEs depending on flexion direction.

## 4. Experiments

In order to verify that CI depends on the mechanical structures, i.e., actuator configuration, segment ratio, overall length, and flexion direction, and ultimately the mechanical structures affect the impact mitigation, a robotic leg was designed and fabricated as shown in Figure 12. The leg consisted of actuators in series with the gear ratio of 11. Note that according to Section 3.1, parallel or serial configurations had no difference in terms of impact. The segment ratio of the upper and lower limbs and the overall length were designed to be almost one and 0.667 m, respectively. The design parameters for the leg are listed in Table 1.

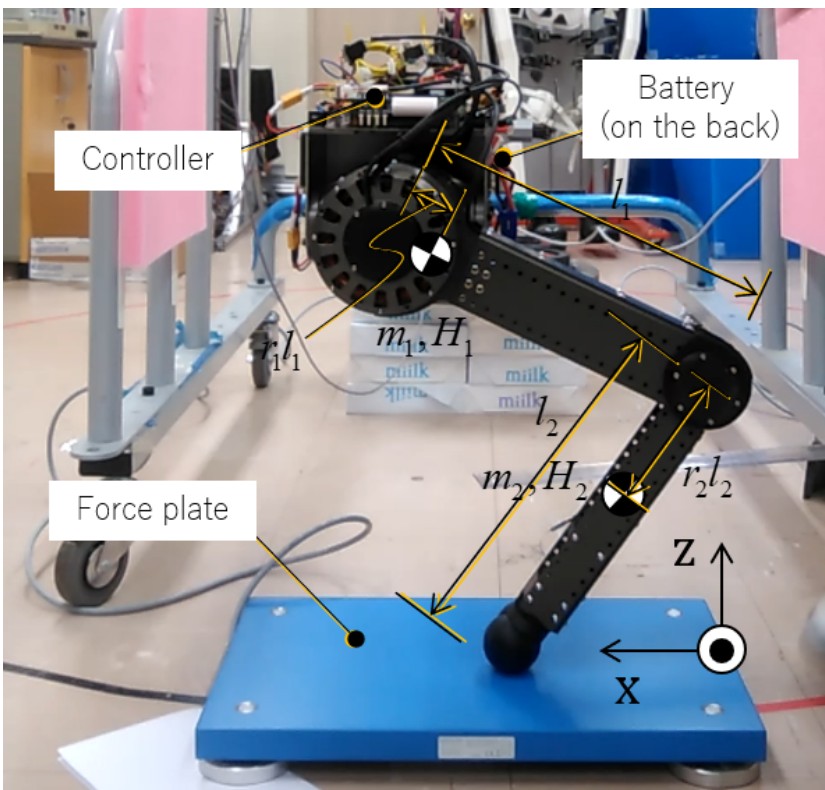

**Figure 12.** Experimental setup.

**Table 1.** Design parameters of a robotic leg.

| Symbol | Unit | Value |
|--------|------|-------|
| $m_1$ | kg | 4.12 |
| $m_2$ | kg | 0.964 |
| $H_1$ | kgm$^2$ | 0.0268 |
| $H_2$ | kgm$^2$ | 0.0363 |
| $l_1$ | m | 0.332 |
| $l_2$ | m | 0.335 |
| $\gamma_1$ | – | 0.0711 |
| $\gamma_2$ | – | 0.331 |

Consider two cases: (a) the leg extends vertically, i.e., $l_n = 0.75$ and $\theta = 0°$, and (b) the leg extends at a angle, i.e., $l_n = 0.75$ and $\theta = 8°$. The reason why the extension ratio is relatively high is because the impacts often occur when the leg extends enough while running; this sufficient extension allows the leg to bear the load after the initial contact. Since the flexion direction was easily changeable compared to other mechanical structures, CIEs of the backward and forward flexions were simulated for the two cases first. Then, CIs were extracted for the limited impact directions as discussed in Section 3.4, as shown in Figure 13. Since the magnitude of the impact direction vector of $\mathbf{v}$ was always 1, therefore, the dimension of $\mathbf{v}$ could be reduced to one, i.e., $\mathbf{v}$ could be re-expressed as $\angle\mathbf{v}$. If $\mathbf{v} = \mathbf{v}_D$, $\angle\mathbf{v} = 0°$, and if $\mathbf{v} = \mathbf{v}_H$, $\angle\mathbf{v} = 90°$. For the two cases, three experimental sets were chosen: (1) and (2) where the leg lands vertically and at an angle with the forward velocity and (3) where the leg lands at an angle only with the downward velocity. According to the simulations, the CI of backward flexion should be larger than that of forward flexion in the sets of (1) and (2). In the set of (3), however, CI of forward flexion should be larger.

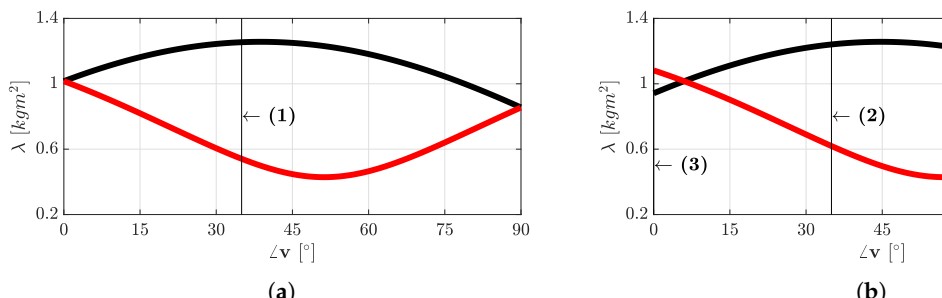

(a)    (b)

**Figure 13.** CI ($\lambda$) for all possible impact directions in the case of (**a**) $l_n = 0.75$ and $\theta = 0°$ and (**b**) $l_n = 0.75$ and $\theta = 8°$; bold black and red represent backward and forward flexions, respectively; intersections where the red and black lines meet the lines marked as (1), (2), and (3) denote the sets of experiments.

### 4.1. Experimental Setup

The impact experiment was set up with the leg as shown in Figure 12. The force plate, the same model used in Section 2.2.3, was used to measure the impact force. Since CI could not be measured directly, the impact forces were measured. As discussed in (17), CI should be dominant in the impact force, and thus, the impact forces could be represented by CIs. In order to control the extension ratio and attack angle of the leg, sbRIO-9651 (*National Instrument Co.*) and GSOLTWIR50/60 (*Elmo Motion Control Ltd.*) were used as the main controller and the motor controller, respectively. Throughout the experiments, the virtual stiffness control was adopted by maintaining the stiffness at 2600 N/m between the hip and the tip of the leg and at 300 N/m and 30 Ns/m in the horizontal direction.

### 4.2. Impact Experiment

As described in Figure 13, the three sets of experiments are as follows: (1) backward/forward flexion with $l_n = 0.75$, $\theta = 0°$, and $\angle\mathbf{v} =\sim 30°$, (2) backward/forward flexion with $l_n = 0.75$, $\theta = 8°$, and $\angle\mathbf{v} =\sim 30°$, and (3) backward/forward flexion with $l_n = 0.75$, $\theta = 8°$, and $\angle\mathbf{v} =\sim 0°$. In the experiments, the virtual stiffness of the leg was regulated to the given extension ratio and attack angle. In the first and second sets, where the attack angle was regulated to $0°$ and $8°$, respectively, the leg was thrown at about 1 m/s from 0.2 m high, which would reach 2 m/s downward when in contact with the ground, such that the impact direction could be estimated around $30°$, as shown in Figure 14a,b. In the third set, the leg was just dropped from the same height such that only the downward velocity was generated, as shown in Figure 14c. In each set, the flexion direction was controlled backward and forward, and each set was repeated 20 times.

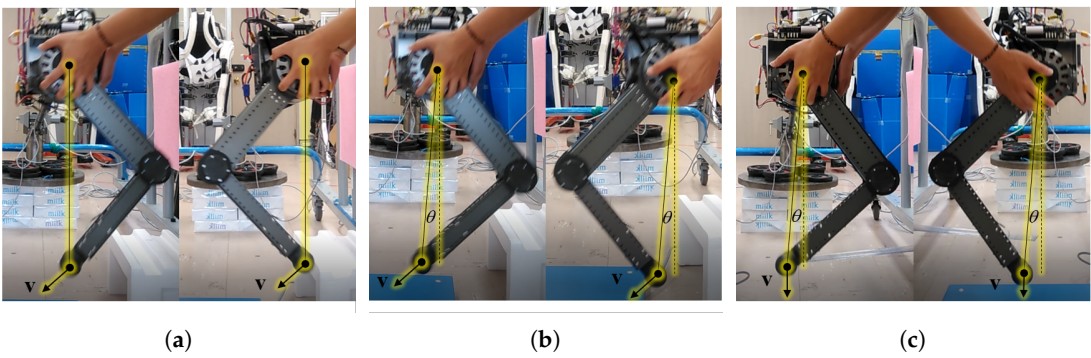

(**a**)                                        (**b**)                                        (**c**)

**Figure 14.** Impact experiment for backward/forward flexion: (**a**) $l_n = 0.75$, $\theta = 0°$, and $\angle\mathbf{v} =\sim 30°$, (**b**) $l_n = 0.75$, $\theta = 8°$, and $\angle\mathbf{v} =\sim 30°$, and (**c**) $l_n = 0.75$, $\theta = 8°$, and $\angle\mathbf{v} =\sim 0°$.

The first peak forces were logged and averaged as shown in Figure 15, where the squares represent the experimental data while the bold and dotted lines represent the mean and standard deviation of the data, respectively. In the sets of (1) and (2), the peak forces of backward flexion were higher than those of the forward flexion, and in the set of (3), the peak forces of forward flexion were higher than those of the backward flexion. This corresponded to Figure 13 because CI of backward flexion was larger than that of forward flexion in (1) and (2), and CI of forward flexion was larger than that of backward flexion in (3).

The experimental result could be organized as in Table 2. In the third and fourth row, the average net force in $x$- and $z$-directions, calculated from the measurements, are shown. In the fifth row, the average net force was derived by averaging the magnitudes of the vectors that consisted of the two forces, i.e., the square root of the sum of squares of the $x$- and $z$-forces. Again, the impact force, described as the average net force in the experiment, was dominated by CI and thus proportional to CI, as discussed in (17). Therefore, the experimental relative CI (rCI) in the sixth row was calculated by dividing the average net force of forward flexion (F) by that of backward flexion (B) and consequently eliminated the need for a proportional constant. At last, the simulated rCIs were derived by dividing CIs of forward flexion by those of backward flexion such that the experimental ones could be directly compared with the simulated ones. Table 2 shows that the experimental and simulated rCIs matched about 98% and that CI could represent the impact force. The result of each set also showed that mechanical structures affected the impact mitigation because rCI was not unity. The experimental and simulated rCIs in Table 2 are visualized in Figure 16, where hollow circles describe the experimental rCIs marked with the set number and gray lines represent all the simulated rCIs, obtained from the data shown in Figure 13, including the simulated rCIs marked with the set number. Figure 16 reconfirms that the experimental and simulated rCIs were almost identical.

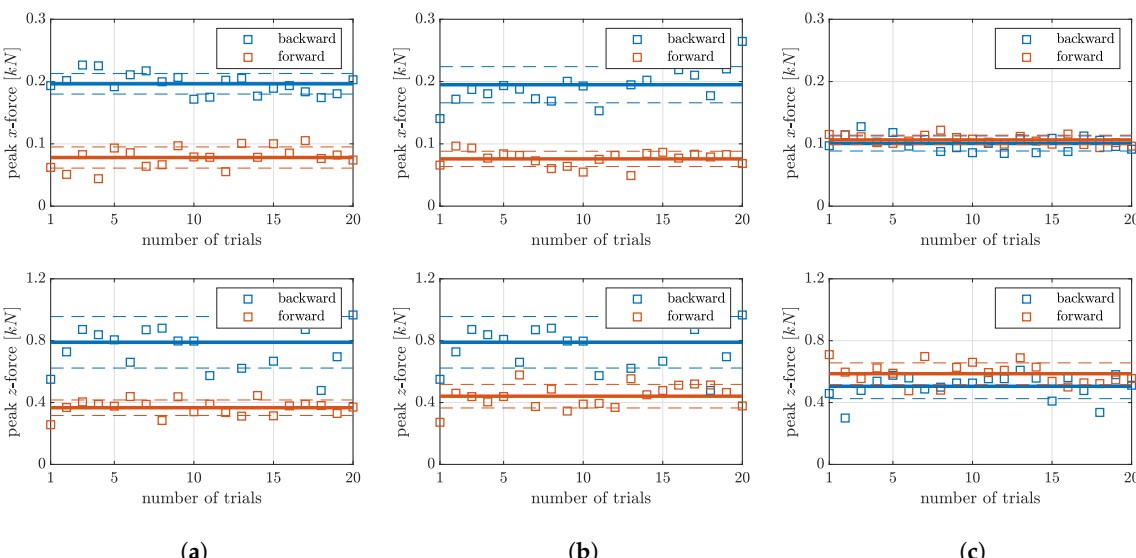

**(a)**    **(b)**    **(c)**

**Figure 15.** Impact forces for backward/forward flexion: (**a**) $l_n = 0.75$, $\theta = 0°$, and $\angle \mathbf{v} =\sim 30°$, (**b**) $l_n = 0.75$, $\theta = 8°$, and $\angle \mathbf{v} =\sim 30°$, (**c**) $l_n = 0.75$, $\theta = 8°$, and $\angle \mathbf{v} =\sim 0°$.

**Table 2.** Impact forces and rCIs for backward (B)/forward (F) flexion.

| Experimental Set | (1) | | (2) | | (3) | |
|---|---|---|---|---|---|---|
| Configuration | B | F | B | F | B | F |
| Average $x$-force (kN) | 0.1964 | 0.0781 | 0.1949 | 0.0759 | 0.1006 | 0.1063 |
| Average $z$-force (kN) | 0.7895 | 0.3664 | 0.7896 | 0.4411 | 0.5050 | 0.5863 |
| Average net force (kN) | 0.8145 | 0.3751 | 0.8141 | 0.4479 | 0.5156 | 0.5960 |
| Experimental rCI | 0.4605 | | 0.5501 | | 1.156 | |
| Simulated rCI | 0.4742 | | 0.5523 | | 1.155 | |

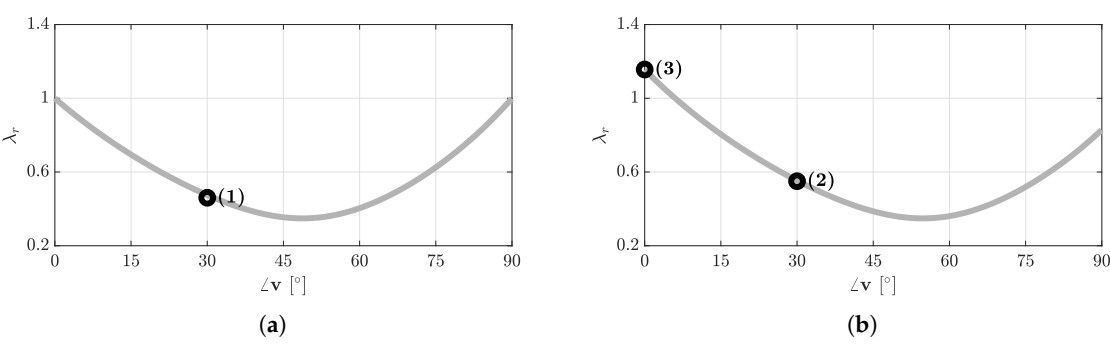

**(a)**    **(b)**

**Figure 16.** rCIs ($\lambda_r$) in the case of (**a**,**b**): hollow circles and gray lines represent the experimental and simulated rCIs, respectively.

## 5. Discussion and Conclusions

This study was motivated by the fact that impact weakens the controllability and durability of the leg of legged robots while running. Since the previous research had been limited to the component-level approach in order to mitigate the impact, the approach needed to move up to the structure-level approach, as proposed. However, in order to evaluate the impact by the mechanical structure, a measure that can quantify the impact was required, and the measure, i.e., CI, could be

derived from the impact dynamics of the leg. Based on the hypothesis that the impact could be represented by CI and in turn, CI was affected by mechanical structures, CI was verified to play a crucial role in the impact and vary on mechanical structures by simulations and experiments.

It could be easily inferred from this paper that smaller CIs were effective for impact mitigation. Since CI varies on the impact direction even if the impact directions were bounded as shown in Figure 11, however, CI, which still can have many different values with respect to a combination of mechanical structures, would not be a good measure of the impact to select mechanical structures effective for impact mitigation. Alternatively, the maximum of CIs can be a good measure of impact. The optimal mechanical structures can be found by minimizing the maximum CI. The average or variance of CIs can also be good measures if the average impact should be mitigated or if the impact is desired such that it should be not sensitive to the impact direction. Depending on the applications, the set range for the possible impact directions can be narrower, wider, or shifted such that the applicable combination of mechanical structures will have more variety.

Interestingly, most legged robots, introduced in Section 1, have their legs flexed backward. This trend might be due to the fact that the forward flexed legs often interfere with stairs or a wide variety of terrains that those robots should overcome. Since the flexion direction was intensively dealt with as one of the mechanical structures in this paper, this study could help to analyze and further improve the mechanical structures of the aforementioned legs in terms of impact.

Technically, running legs require two more properties besides effective impact mitigation: great force production and fast swing recovery. Although the paper focused on the effect of mechanical structures on impact mitigation, the mechanical structures are also supposed to affect force production and swing speed, as previously investigated in [26]. Accordingly, this study, which focused on impact mitigation only, should be extended to define measures to evaluate those three properties and, thus, to comprehensively analyze the mechanical structures, not individually. A subsequent paper will deal with the analysis in the near future.

**Author Contributions:** Conceptualization, J.C. and K.K.; methodology, J.C.; software, J.C.; validation, J.C.; formal analysis, J.C.; investigation, J.C.; resources, K.K.; data curation, J.C.; writing, original draft preparation, J.C.; writing, review and editing, J.C. and K.K.; visualization, J.C.; supervision, K.K.; project administration, K.K.; funding acquisition, K.K. All authors read and agreed to the published version of the manuscript.

**Funding:** This research was funded by the Korea Evaluation Institute of Industrial Technology (KEIT) Grant Number 10080320.

**Acknowledgments:** The authors gratefully acknowledge robot components fabricated by Woo at Baekdu Engineering and the experimental site provided from KAIST.

**Conflicts of Interest:** The authors declare no conflict of interest. The funders had no role in the design of the study; in the collection, analyses, or interpretation of data; in the writing of the manuscript; nor in the decision to publish the results.

## Abbreviations

The following abbreviations are used in this manuscript:

| | |
|---|---|
| MIT | Massachusetts Institute of Technology |
| UCLA | University of California, Los Angeles |
| ETH | Eidgenössische Technische Hochschule Zürich |
| KAIST | Korea Advanced Institute of Science and Technology |
| COM | Center Of Mass |
| CIE | Contact Inertia Ellipse |
| CIM | Contact Inertia Matrix |
| CI | Contact Inertia |
| rCI | Relative Contact Inertia |

## Appendix A. Impact Dynamics of an Articulated Rigid Body

*Appendix A.1. Impact Dynamics*

When an articulated rigid body, comprised of a body and upper and lower limbs, impacts the ground as shown in Figure A1, its dynamics is expressed as in (16), where:

$$\mathbf{H}(\mathbf{q}) = \begin{bmatrix} \mathbf{H}_{11} & \mathbf{H}_{12} \\ \mathbf{H}_{21} & \mathbf{H}_{22} \end{bmatrix}, \tag{A1}$$

$$\mathbf{H}_{11} = \begin{bmatrix} m_0 + m_1 + m_2 & 0 \\ 0 & m_0 + m_1 + m_2 \end{bmatrix},$$

$$\mathbf{H}_{12} = \begin{bmatrix} -m_{12}l_1 c\theta_1 - m_2 r_2 l_2 c\theta_{12} & -m_2 r_2 l_2 c\theta_{12} \\ m_{12}l_1 s\theta_1 + m_2 r_2 l_2 s\theta_{12} & m_2 r_2 l_2 s\theta_{12} \end{bmatrix},$$

$$\mathbf{H}_{21} = \begin{bmatrix} -m_{12}l_2 c\theta_1 - m_2 r_2 l_2 c\theta_{12} & m_{12}l_2 s\theta_1 + m_2 r_2 l_2 s\theta_{12} \\ -m_2 r_2 l_2 c\theta_{12} & m_2 r_2 l_2 s\theta_{12} \end{bmatrix},$$

$$\mathbf{H}_{22} = \begin{bmatrix} h_{11} & h_{12} \\ h_{12} & h_{22} \end{bmatrix}$$

where $h_{11} = H_1 + H_2 + \left(m_1 r_1^2 + m_2\right) l_1^2 + m_2 r_2^2 l_2^2 + 2 m_2 r_2 l_1 l_2 c\theta_2$, $h_{12} = H_2 + m_2 r_2^2 l_2^2 + m_2 r_2 l_1 l_2 c\theta_2$, and $h_{22} = H_2 + m_2 r_2^2 l_2^2$,

$$\mathbf{B}(\mathbf{q}, \dot{\mathbf{q}}) = \begin{bmatrix} m_{12}l_1 s\theta_1 \dot{\theta}_1^2 + m_2 r_2 l_2 s\theta_{12} \dot{\theta}_{12}^2 \\ m_{12}l_1 c\theta_1 \dot{\theta}_1^2 + m_2 r_2 l_2 c\theta_{12} \dot{\theta}_{12}^2 \\ -m_2 r_2 l_1 l_2 s\theta_2 \dot{\theta}_{12}^2 + m_2 r_2 l_1 l_2 s\theta_2 \dot{\theta}_1^2 \\ m_2 r_2 l_1 l_2 s\theta_2 \dot{\theta}_1^2 \end{bmatrix}, \tag{A2}$$

$$\mathbf{G}(\mathbf{q}) = \begin{bmatrix} 0 \\ m_O + m_1 + m_2 \\ (m_1 + m_2) l_1 s\theta_1 + m_2 r_2 l_2 s\theta_{12} \\ m_2 r_2 l_2 s\theta_{12} \end{bmatrix}, \tag{A3}$$

and:

$$\mathbf{T} = \begin{bmatrix} 0 \\ 0 \\ \tau_1 + \tau_2 \\ \tau_2 \end{bmatrix} \tag{A4}$$

In (A1), (A2), and (A3), $m_{12}$, $\theta_{12}$, $s\theta$, and $c\theta$ are short for $m_1 r_1 + m_2$, $\theta_1 + \theta_2$, $\sin\theta$, and $\cos\theta$, respectively. In the same manner as in (4), let:

$$\bar{\mathbf{J}}(\mathbf{q}) = \begin{bmatrix} \mathbf{I}_{2\times 2} & \mathbf{J}(\mathbf{q}) \end{bmatrix} \in \mathbb{R}^{2\times 4}, \tag{A5}$$

where:

$$\mathbf{J}(\mathbf{q}) = \begin{bmatrix} \frac{\partial \mathbf{R}(\theta_1)}{\partial \theta_1} \hat{l}_1 + \frac{\partial \mathbf{R}(\theta_{12})}{\partial \theta_{12}} \hat{l}_2 & \frac{\partial \mathbf{R}(\theta_{12})}{\partial \theta_{12}} \hat{l}_2 \end{bmatrix} \in \mathbb{R}^{2\times 2}. \tag{A6}$$

Through a similar process from (5) and (6) to (7), the impact dynamics of an articulated rigid body can be derived as in (17).

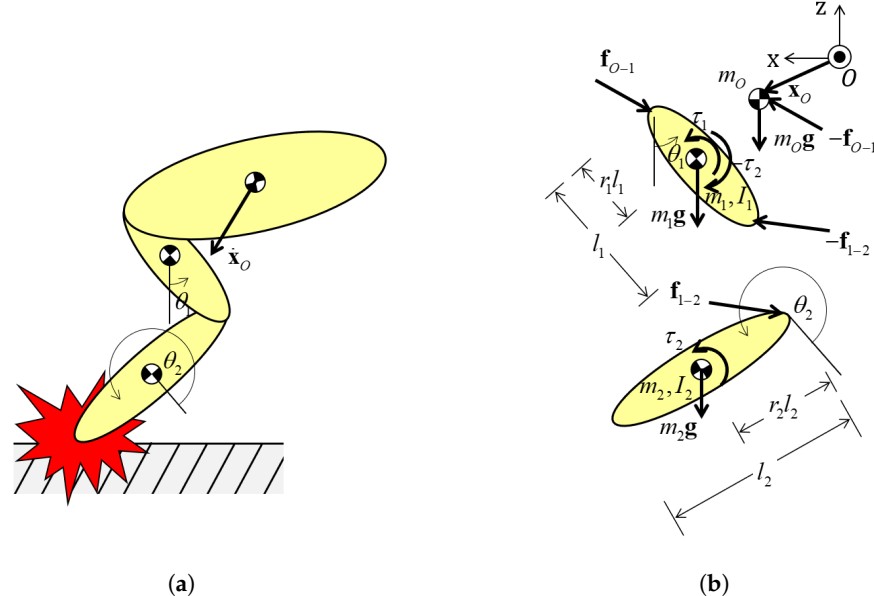

**Figure A1.** Articulated rigid body dynamics: (**a**) impact; (**b**) free body diagram.

*Appendix A.2. Reduced form of Contact Inertia*

By the *LDU* decomposition, $\mathbf{H}^{-1}$ is decomposed into the following.

$$\mathbf{H}^{-1} = (\mathbf{LDU})^{-1} \tag{A7}$$

where:

$$\mathbf{D}^{-1} = \begin{bmatrix} \mathbf{H}_{11}^{-1} & \mathbf{0}_{2\times2} \\ \mathbf{0}_{2\times2} & \left(\mathbf{H}_{22} - \mathbf{H}_{21}\mathbf{H}_{11}^{-1}\mathbf{H}_{12}\right)^{-1} \end{bmatrix} \in \mathbb{R}^{4\times4}, \tag{A8}$$

$$\mathbf{L}^{-1} = \begin{bmatrix} \mathbf{I}_{2\times2} & -\mathbf{H}_{11}^{-1}\mathbf{H}_{12} \\ \mathbf{0}_{2\times2} & \mathbf{I}_{2\times2} \end{bmatrix} \in \mathbb{R}^{4\times4}, \text{ and} \tag{A9}$$

$$\mathbf{U}^{-1} = \begin{bmatrix} \mathbf{I}_{2\times2} & \mathbf{0}_{2\times2} \\ -\mathbf{H}_{21}\mathbf{H}_{11}^{-1} & \mathbf{I}_{2\times2} \end{bmatrix} \in \mathbb{R}^{4\times4}. \tag{A10}$$

Since $\mathbf{H}_{11}$ is considered much larger than $\mathbf{H}_{12}$, $\mathbf{H}_{21}$, and $\mathbf{H}_{22}$ due to the body mass of $m_O$ included, all the terms with the product of $\mathbf{H}_{11}^{-1}$ can be assumed negligible, and thus, the nullity of (A8) becomes two while (A9) and (A10) become identity matrices. Consequently, (18) can be reduced to (19).

**Appendix B. Contact Inertia: Serial and Parallel Actuator Configuration**

Let the inertia and Jacobian matrices from the serial and parallel actuator configurations:

$$\mathbf{H}_{22(s)} = \begin{bmatrix} x+a+2b & a+b \\ a+b & a \end{bmatrix}, \quad \mathbf{H}_{22(p)} = \begin{bmatrix} y & b \\ b & a \end{bmatrix} \tag{A11}$$

and:

$$\mathbf{J}_{(s)}(\mathbf{q}) = \begin{bmatrix} c+e & e \\ d+f & f \end{bmatrix}, \quad \mathbf{J}_{(p)}(\mathbf{q}) = \begin{bmatrix} c & e \\ d & f \end{bmatrix}, \tag{A12}$$

where the serial and parallel configurations are subscripted as $(s)$ and $(p)$, respectively. Elements in the matrices of (A11) and (A12) are listed below:

$$x = H_{1(s)} + \left[ (m_1 + M) \, r_{1(s)}^2 + m_2 \right] l_1^2, \tag{A13}$$

$$y = H_{1(p)} + \left( m_1 r_{1(p)}^2 + m_2 \right) l_1^2, \tag{A14}$$

$$a = H_2 + m_2 r_2^2 l_2^2, \tag{A15}$$

$$b = m_2 r_2 l_1 l_2 \cos \theta_2, \tag{A16}$$

$$c = -l_1 \cos \theta_1, \tag{A17}$$

$$d = l_1 \sin \theta_1, \tag{A18}$$

$$e = -l_2 \cos (\theta_1 + \theta_2), \tag{A19}$$

and:

$$f = l_2 \sin (\theta_1 + \theta_2). \tag{A20}$$

Applying (A11) and (A12) to (19),

$$\Lambda = \frac{1}{xa - b^2} \begin{bmatrix} k_1(x) & k_2(x) \\ k_3(x) & k_4(x) \end{bmatrix} \tag{A21}$$

for the serial configuration and:

$$\Lambda = \frac{1}{ya - b^2} \begin{bmatrix} k_1(y) & k_2(y) \\ k_3(y) & k_4(y) \end{bmatrix} \tag{A22}$$

for the parallel configuration, where $k_1(\gamma) = (ca - eb) c + (e\gamma - cb) e$, $k_2(\gamma) = (ca - eb) d + (e\gamma - cb) f$, $k_3(\gamma) = (da - fb) c + (f\gamma - db) e$, and $k_4(\gamma) = (da - fb) d + (f\gamma - db) f$. $x$ and $y$ represent the moments of inertia about the upper joint axis for the upper limb. Applying (21) and (20) to (A13) and (A14), it will be clear that $x$ and $y$ are equal. Therefore, CIs from the serial and parallel actuator configuration are equal.

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
