# Peer review of "The Analysis of Mechanical Structure of a Robotic Leg in Running for Impact Mitigation"

_applsci, doi:10.3390/app10041365_

Round 1

Reviewer 1 Report

The article addresses a current topic of great interest in the study of walking robots, analyzing how different constructive factors affect the impact forces of the legs against the ground during the march

But the article needs substantial changes to be published

On the one hand, the structure of the article is confusing and infrequent in an applied engineering article. The organization in Introduction, Results, Discussion and Materials and Methods makes reading and understanding difficult. It is considered more appropriate for a correct understanding of work to adapt to the scheme of Abstract, Introduction, Formalization of the problem, Methods and experiments, Results, Discussion and Conclusions
It is noteworthy that the article does not present Conclusions

On the other hand, sections with mathematical development are lacking in detail, omitting steps that require excessive effort by the reader to understand the argumentation and obtain results. The notation used in some cases and the lack of definition of a considerable part of the variables involved in the expressions does not help either.

Some specific aspects found in the paper are detailed below:

Figure 1: the distance indicated for r1(s).l1 seems not to be correct

Equation (4): Define all the variables and parameters that appear in the equation (R, theta, l1, l2, ...) This is applicable to all equations. Every term in the equations must be clearly defined in the text. Do not assume that the reader can deduct it

Line 117: The angle of the velocity vector appears using a /_xdot symbol that must be defined. Not doing so makes reading difficult and slows down.

Line 122: The Force Plate model used is not indicated (it is not indicated until line 225). It should be indicated here, for consistency with the information on the controllers (lines 129 and 130).

Figure 7: What is Lambda (λ)? Again a term without a previous definition

Table 2: I do not understand how the value of Average net Force is obtained. Can you explain it?

Table 2: What are the differences between the values given to the Relative CI and those obtained from F / B? For example, for case (3) F / B = 0.25 * 0.22 = 1,136, while 1.11 appears in the table.

Line 172: Again two expressions are used without explaining their meaning: XF_dot and XD_dot

Figure 11: It is NOT indicated that it is Lambda _uppercase (Λ)

Section 4.1.2 and 4.2.2: They must be rewritten clarifying all developments. The lack of details in the deductions makes the reading complicated and confusing. The notation used does not facilitate it. For example:
(7): appears H (bold) and H (without indicating that it is this second)
(8): I is "Impulse" or the Identity matrix?
Line after (8): (l circumflex) l^ = [0 -l] should not be transposed ([0 -1]T?
(11): H should not be the inverse? Λ-1 = J.H-1 .JT. (as in (21))
(13): It must be explained how the expression is deduced

4.2.3 Since the experiment is performed without any control (“dropped 10 times repeatedly from about 0.5m”; “dropping it with an angle of about 30”), how is it possible to validate the results? It is surprising that the experiment with an angle of approximately 30 degrees has less standard-deviation than the vertical drop, when this (perpendicular to the ground) is more repetitive.

Author Response

Please check out the attachment.

Reviewer 2 Report

In this paper, leg mechanisms are developed to effectively mitigate the impact due to the consistent collisions with the ground and discussed what is a good measure for impact mitigation and additional consideration of leg mechanisms. After the correction of the problems below, this paper could be considered for publication.

There are some recent developments in the field of robotics using machine learning algorithms to guarantee controllability and durability. I suggest the author add some sentences of literature review to make the introduction section more adequate. For example:
@article{li2018enhanced,
title={An enhanced teaching interface for a robot using DMP and GMR},
author={Li, Chunxu and Yang, Chenguang and Ju, Zhaojie and Annamalai, Andy SK},
journal={International journal of intelligent robotics and applications},
volume={2},
number={1},
pages={110--121},
year={2018},
publisher={Springer}
}

The paper is well written, but the contributions or novelties of this work are not clear. The paper reads like a technical report. For a journal paper, new
methods or experimental results must be included.

The whole structure of the paper looks unclear, the authors should better organize.

The strategy remains complex to implement with many parameters. A discussion on the adjustment of these parameters should be presented in the document.

There are some minor typo errors, please check carefully.

Author Response

Please check out the attachment.

Reviewer 3 Report

This paper investigate how to mitigate the impact force, the actuator configuration, segment ratio, total length, and flexion direction and contact inertia (CI) defined at the tip of the leg is derived and utilized to analyze the leg mechanisms in terms of impact mitigation. 

Some of the suggestions

Could you do some simulation to simulate what is the contact inertia and impact force in different mass, length, flexion direction, and segment ratio configuration. I think it will let the reader to understand how to chose those configuration as they design a leg. It will be better to have a table to review your developed leg and other legs you mention in the introduction. In the table, you can compare those parameter you discuss in this papers. It will be let us understand what are the state-of-art legs and what is their configuration of leg. It seems like reducing the CI can mitigate the impact force. But how to reduce the CI ? Could you summary how to reduce CI in detail.   

Author Response

Please check out the attachment.

Round 2

Reviewer 1 Report

Reviewer #1 Table 2: What are the di erences between the values given to
the Relative CI and those obtained from F / B? For example, for case (3) F /
B = 0.25 * 0.22 = 1,136, while 1.11 appears in the table.
Authors Authors appreciate your comment. Thanks to your comment, authors
found errors in numbers and gures. Along with Fig. 16, TABLE 2 was cor-
rected with an additional row of \Simulated rCI" to directly compare it with
2
the experimental rCI, which is visualized in Fig. 16. The new term of \rCI" is
de ned in Line 242 and the Abbreviations.

Please , review again the numbers taht appears un Table 2. The errro are really little, but why not use the correct values?

Experimental rCI 0.46 0.55 1.16  --> 0.47 0.56 1.15

These values are calculated as 0.38/0.81  0.45/0.81   0.6/0.52

Author Response

Authors appreciate your comment, and fully understand your concern. Authors found that the difference comes from the truncation error in the process of calculation. Accordingly, authors increase the significant digits from two to four such that the truncation error becomes minor.

Reviewer 2 Report

After the 2nd round of revision, the authors have clearly addressed my concerns.

Author Response

Authors wholeheartedly appreciate your constructive review.